# Optimization strategies for conservation of traditional dwellings in Hongcun Village, China, based on decay phenomena analysis

**Tingshen Li**[1], **Minghao Zhang**[2]*, **Xianguang Gu**[2]

**1** College of Environmental Engineering, Xuzhou University of Technology, Xuzhou, China, **2** School of Architecture and Design, China University of Mining and Technology, Xuzhou, China

* archmz@cumt.edu.cn

## Abstract

Hongcun traditional dwellings are representative of Huizhou architecture in China, with distinctive cultural and architectural characteristics in terms of layout, material, decoration and other features. Research on Hongcun traditional dwellings has been a matter of ongoing interest in academic circles in China, but there has been no specific focus on the phenomena of decay affecting these structures, even though research on this aspect has the most direct impact on the conservation of traditional dwellings. In this study, abundant and comprehensive fieldwork was carried out to investigate the building information, materials and especially preservation status of traditional dwellings. Furthermore, the decay phenomena of traditional dwellings were identified and described in detail in the Masonry Components and Wooden Components sections, which are based on the collected information and the relevant guidelines. Moreover, the restoration and actual conservation practices for traditional dwellings, which were specifically both government-led and private projects, were examined. In these analyses, the main problems related to the decay phenomena investigation and intervention are systematically summarized, and corresponding solutions are proposed to ensure that optimized conservation strategies are applied to traditional dwellings in Hongcun village.

## 1. Introduction

Hongcun is a traditional village in Yi County, southern Anhui Province, China (*30°11' N, 117° 38' E*), that was established in 1131 and thus has a continuous history of over 870 years. The site covers 19.11 hectares and lies at the foot of Lei Gang Hill, which is in close proximity to Huangshan Mountain and Yangtze River, and has a subtropical warm monsoon climate, with an annual average temperature of approximately 15˚C (see Fig 1). The traditional dwellings of Hongcun are representative of Huizhou region traditional dwellings, which have important regional cultural significance [1]. Huizhou region is located in the southern Anhui Province, China, and has a recorded history of more than 2200 years. As this region has rich and distinct cultural (and architectural) characteristics, the State Council approved the establishment of

**Data Availability Statement:** All data, models, and code generated or used during the study appear in the submitted article.

**Funding:** Xuzhou Science and Technology Plan Project Grant Number: KC21305 Awarded to: Prof. Minghao Zhang The funder had no role in study design, data collection and analysis, decision to publish, or preparation of the manuscript.

**Competing interests:** NO authors have competing interests

Huangshan City in 1987 in order to strengthen the tourism development of Huangshan Mountain. As a consequence, the Huizhou region disappeared from the administrative divisions of China. Because of the area's local geography, economic conditions, social structural particularities and self-regulated development, the traditional dwellings of Hongcun still retain the original patterns and historical characteristics of the Ming and Qing dynasties in a state of relative completeness, and the people of Hongcun continue to live within their traditional dwellings [2].

Since the late 1980s, the Chinese academic circles represented by Prof. Shan Deqi [3] have successively carried out research on the history and culture of traditional dwellings in the Huizhou region. In 2000, Hongcun was designated a UNESCO World Heritage Site together with Xidi Village due to the outstanding state of preservation of their street plans, architecture and decorative elements and the unique integration of the architecture and water management systems [4]. Since Hongcun's inclusion on the World Heritage List, the regulations, documentation and research on its traditional dwellings have not reflected a high degree of epistemological continuity, presenting distinct goals and approaches at various stages [5, 6]. The Conservation Plan refers to the overall plan for a heritage site, including both conservation and management. At present, four versions of the specialized Conservation Plan for Hongcun have been released; however, these submissions focused on anthropological studies and strategies for utilizing the traditional dwellings rather than on investigating the existing materials and the actual reasons for decay phenomena or proposing targeted conservation strategies, a situation that greatly affects the conservation of Hongcun's authenticity [7, 8].

In recent years, some scholars have begun to gradually conduct research on the architectural materials of Hongcun traditional dwellings. Liu [9] analyzed the construction process by studying the main materials used in Hongcun traditional dwellings. Liu [10] focused on investigating vernacular materials in Hongcun in terms of color, shape, texture, etc. to explore their transformation and application patterns. Chang et al. [11] focused on understanding the unique ecological characteristics of Hongcun traditional dwellings through comprehensive research on plaster and wood. These projects provided important basic information about Hongcun traditional dwellings for this study, but the vernacular materials were not systematically classified and examined, so they could not guide the further analysis and diagnosis of decay phenomena.

In terms of decay analysis of and interventions in traditional dwellings, Musso et al. [12] investigated the rural architectural heritage of Cinque Terre, describing in detail the decay of each building component and proposing specific conservation and restoration measures. Giambruno et al. [13] analyzed traditional buildings in Multan, Pakistan in terms of brick, stone, wood and other material aspects and comprehensively clarified the decay phenomena. Liu et al. [14] conducted an in-depth study on the influence of salt efflorescence on traditional brick buildings from the perspective of meteorological parameter analysis. Qin [15] combined specific practical cases to analyze the decay of traditional brick-wood buildings, such as case of insect pests and rotten wooden components as well as the efflorescence and cracking of brick components, and formulated relevant points of governance and intervention suggestions. Chen et al. [16] collected information on different humidity decay phenomena of vernacular dwellings and verified the pathological formation mechanism. These researchers made attempts to classify and intervene in decay problems, especially in dividing the materials of traditional dwellings into mainly wood and masonry categories, which had significant reference value for this study.

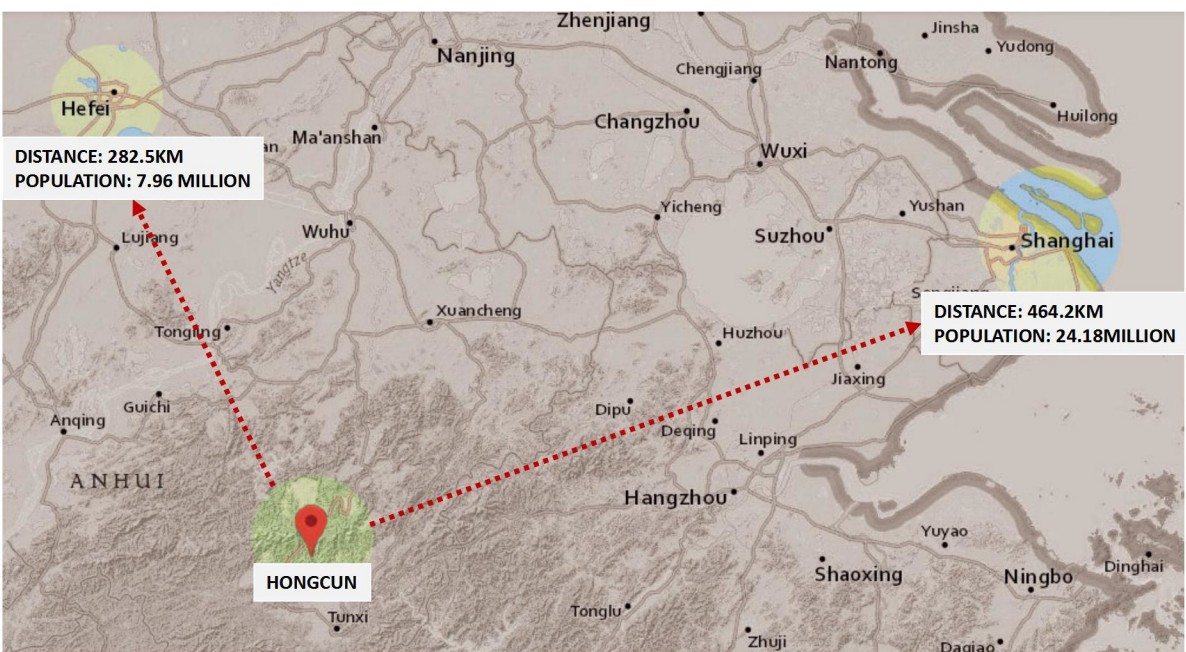

**Fig 1. The site location of Hongcun and the distances to major cities.** (The figure is obtained from the USGS National Map Viewer). Source: Annotated by author.

## 2. Research significance and methods

According to the literature, many studies have been conducted to investigate and analyze the materials and decay phenomena of traditional dwellings, but the relevant research specifically on Hongcun traditional dwellings is comparatively limited, especially regarding lack of basic research data on architectural materials, present status and decay problems; thus, previous studies have not provided the fundamental data needed for the proposal of conservation measures. In addition, in research on interventions in architectural decay, the literature provides a variety of research methods and achievements, but due to the unique environment and construction techniques in Hongcun, these conservation measures are not necessarily compatible with the traditional dwellings there, and the lack of analysis of actual conservation practices, makes it impossible to determine the suitability of conservation measures.

Therefore, this study was undertaken with a standard methodological approach, combining physical observations and documentation of the traditional dwellings with an ethnographic study of the local community that entailed conducting semistructured interviews, recording oral histories and examining the secondary literature. According to the latest Conservation Plan for Hongcun (2016–2030), there are still 110 traditional dwellings dating to the Ming and Qing Dynasties, which is the period under study in this paper. This study consisted of two steps. In the first step, extensive field investigations were carried out on the traditional dwellings in Hongcun, and the building information, materials (wood frame, masonry, lime mortar, plaster coating, etc.) and especially decay phenomena (masonry components and wooden components) of the dwellings were comprehensively investigated and analyzed in depth. In the second step, specific cases of government-led and private conservation projects of Hongcun traditional dwellings were selected, actual problems encountered in the intervention process were evaluated, and strategies to optimize the conservation measures were proposed in combination with the analytical and diagnostic results of material decays. Fig 2 shows the general overview of this study.

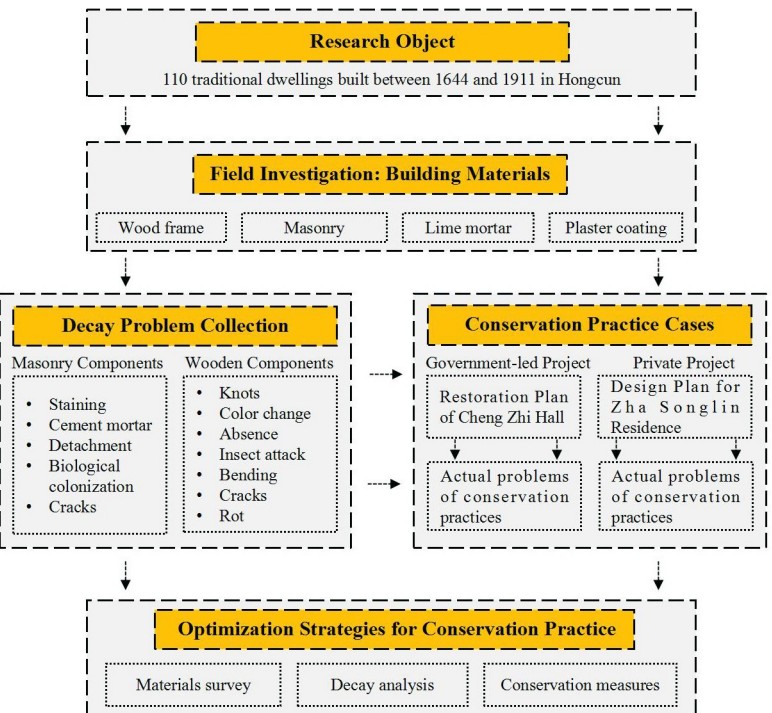

**Fig 2. Schematic of the research program followed in this study.** Source: Drawn by author.

## 3. Materials

The geographical surroundings of Hongcun are hilly, and good building ground is relatively scarce; therefore, two-floor or three-floor brick and wood buildings are constructed rather than the single-level courtyard houses more commonly seen elsewhere [17]. In Huizhou, the architecture has traditionally strictly followed the ancient principles of Feng Shui, such that the way in which the village is situated relative to a hill and/or hydrological features determines the accumulation of life force, or QI, which in a material sense translates into numerous off-spring and a prosperous village [18]. As one might expect, the layout and architecture of Hongcun also follow the principles of Feng Shui, vividly showing the unique characteristics of the Huizhou settlement form and space [19].

Around the beginning of the 15[th] century, in accordance with the instructions of Feng Shui master, the founder of Hongcun, Wang Siqi, directed the villagers to extend the natural spring to form a semicircular pond, which is called Moon Pond and forms the core of the village. They then dug the so-called *shui zhen* (the ditches), water channels that converge in Moon Pond, thus forming the primary hydraulic system of Hongcun, providing villagers with water for drinking, washing and fire protection [20]. While Moon Pond and the *shui zhen* significantly improved fire protection, the elders were still worried about the danger of forest fires. Therefore, in 1607, the 81[st] Patriarch Wang Kuiguang led the excavation of South Lake over three years, after which the entire pattern and water system of Hongcun were essentially complete [21].

During the late Ming and early Qing Dynasties (approximately 1627 to 1684), a new development in the typical "three-room with corridor" pattern gradually appeared. This new pattern was not drastically different from the pattern of the existing traditional dwellings within Hongcun. In contrast to the earlier buildings, the present traditional dwellings in Hongcun are frame structures, and the load-bearing structures are wooden frame systems, while the exterior

wall is just an enclosure structure, which means that the exterior wall only has the function of enclosing the traditional dwelling and does not bear other loads. Therefore, the materials and techniques used for these exterior walls are relatively flexible.

As most of the male population of Hongcun was traditionally engaged in business away from their hometown, security for their families was provided by the 7–8 m enclosure walls, designed in three separate vertical sections of stone in order to avoid basal erosion from the *shui zhen* water channel. The foundation materials are usually constructed of gravel, granite or a mixed base. A granite base has better integrity, and a gravel foundation is easy to transport and construct. Therefore, the practice of the mixed base is usually adopted, as it can not only provide good integrity, but also reduce the pressures of transportation and construction and combine the advantages of both.

In the middle section, local black brick is the typical exterior facing material. It is fired with clay, and in the firing process, the oxidized $Fe^{3+}$ fraction is reduced to $Fe^{2+}$, so the appearance is blue. The empty bucket masonry method was used in construction, as the wall is not load bearing. The hollow portion is filled with yellow slurry, as the local soil of Hongcun is acidic yellow or red soil, and can be used as a bonding material to fill empty bucket walls. After the sand is washed with yellow mud to make it sticky, lime is mixed in to obtain yellow slurry. The lime is often used in wall plastering, creating a white wall, which is an important element of traditional dwellings in Hongcun [22].

In the upper section, the gable walls are constructed with gray tiles in a stepped or ridge shape that looks like a horse head; therefore, the gable walls are often also called *horse-head walls*. In the case of a fire in the neighborhood, the *horse-head walls* serve as a fire proofing method; thus, over time, these *horse-head walls* became an architectural element specific to Hongcun's traditional dwellings (see Fig 3).

Additionally, Chinese traditional dwellings are made of wooden beam column frame structures, and the traditional dwellings at Hongcun are no exception. For the wooden materials of Hongcun traditional dwellings, pine, fir and ginkgo trees are usually chosen. The pillars are first set in the ground and the girders are erected over the pillars, with the roof being lain over the girders, which causes all the weight of the roof to be directed to the girders, then to the pillars and finally to the ground. The main beam is commonly known as the *White Gourd Beam* and is usually adorned with exquisite wood carvings. A skylight located next to the hall is located approximately 50cm above the other beams in the hall so that lighting from the skylight can penetrate the innermost area and thus expand the depth of the hall [23]. Correspondingly, the hall expands in width and height so that it appears more spacious. Wood carvings are the major interior decorations of the traditional dwellings. In the late Qing Dynasty (around 1851 to 1911), wood carvings were used on crescent beams, horizontal tables, bucket arches, sparrow braces, hangings, lotus doors and window boards, and these carvings are rich and varied in terms of subject matter. They reflect the pure, natural and sincere ethics of the period as well as the practical and aesthetic combination of design concepts.

The above research on the architectural materials reveals that most traditional dwellings in Hongcun are wooden structures supplemented by brick, tile and stone and are made up of an exterior enclosure, a wooden structure and an interior component [24]. Furthering existing studies, the next section identifies and classifies the extant decay phenomena and structural pathologies of the walls and wooden components of the traditional dwellings of Hongcun and presents actual details of their decay. This will aid in the subsequent discussion of conservation practices.

## 4. Decay phenomena of materials

In Hongcun, the main heritages are undoubtedly the traditional dwellings, and the decay and damage these built heritages present is caused not only by time, the natural wear and tear of

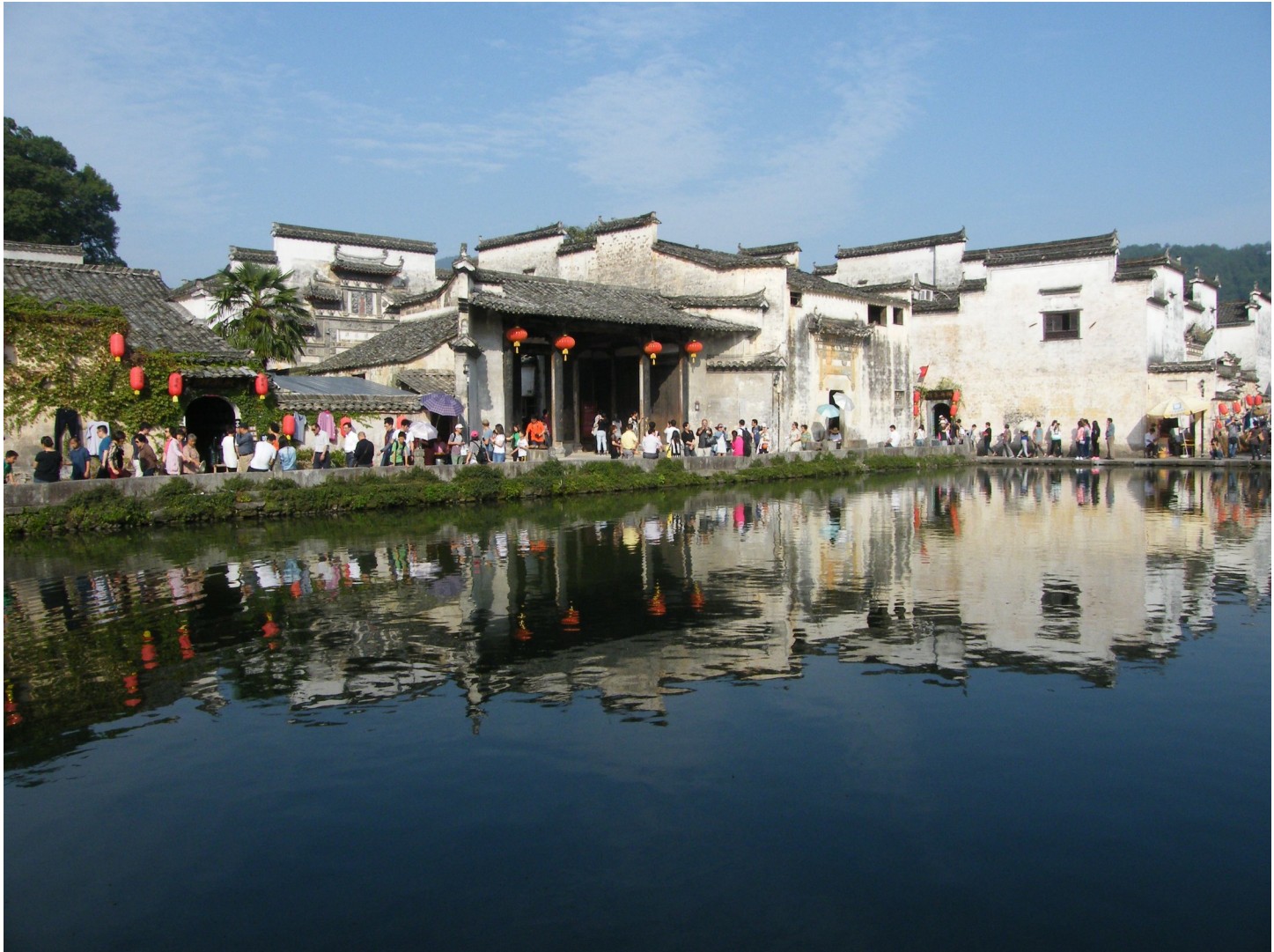

**Fig 3. Moon Pond and the exterior of the traditional dwellings along the pond.** Source: Taken by the author.

use, and accidental disasters but also by moisture penetration, insect pests, *etc*. Indeed, the Conservation Plan 2016 also proposes the relevant articles, such as "Restoration with Present Status" and "Major Restoration", should prepare corresponding restoration plans for specific built heritages. However, there has been a lack of specific analysis and research on the material decay problems of Hongcun traditional dwellings, which has led to a lack of unity in specific explanations and interpretations of the technical standards of the restoration process and an insufficient theoretical basis for specific restoration projects of traditional dwellings. In reality, most decay phenomena can be avoided with essential and basic daily maintenance. Therefore, this section will examine the decay of the traditional dwellings based on their complex present status and the multiple materials and techniques involved in their construction.

Under these circumstances, the analysis of the decay phenomena for architectural materials using international references is essential, and the Illustrated Glossary on Stone Deterioration Patterns by ICOMOS-ISCS covers the general situations of the decay phenomena of stone. A preliminary attempt to analyze the current types and actual situation of the decay phenomena

of the traditional dwellings based on international research and a classification of the materials will be shown below. Although the preservation status of each traditional dwelling is different, the construction methods and their specific decay phenomena can be analyzed in two primary dimensions: the Masonry Components and the Wooden Components (including the wooden structure and the interior component). These together encompass all the major relevant pathologies. No permits were required for the described study, which complied with all relevant regulations.

## 4.1 Identifying and classifying the decay phenomena of Masonry components

Initially, moisture and mildew were some of the main problems affecting the traditional dwellings. The location of Hongcun in a high rainfall region as well as the unique *shui zhen* water system (see above), which creates a pleasant interior and exterior environment, results in high ambient humidity levels. Over the course of many years, this humidity has inevitably affected the traditional dwellings, resulting in a series of problems such as cracks in the walls, mustiness and blacking of the walls, plant growth in the walls, plaster flaking and detachment, thinning of the bonding lime layer of the gravel foundation and, worst of all, erosion, which has, more than any other problem, accelerated the degradation of the masonry components of the traditional dwellings. Additionally, many problems in the masonry structure have been caused by problematic intervention measures during restoration or renovation, particularly the use of cement mortar to fill cracks, which is most commonly the result of insufficient understanding of conservation techniques, international standards and established best practice. Other serious problems, such as holes or cracks or even structural instability causing partial collapse of the wall, are caused by the lack of habitual daily maintenance.

Based on the issues mentioned above, the decay phenomena of the masonry components are caused by many factors. By referring to the document *Illustrated Glossary on Stone Deterioration Patterns* (ICOMOS-ISCS), this study classified the main decay phenomena as Staining, Cement Mortar Fillling, Detachment, Biological Colonization and Cracks based on the investigation of the traditional dwellings in Hongcun (see Fig 4). It listed the main phenomena and causes of the decay and displayed and explained the problems through specific decay photos of the masonry components.

## 4.2 Identifying and classifying the decay phenomena of Wooden components

By nature, the wooden frame structures of Hongcun traditional dwellings are built such that *"a slight move in one part may affect the situation as a whole"*. Frequently, when a girder, beam or pillar is damaged and needs to be repaired, the trabeculae, pillarets and other related components all need to be replaced, which requires a relatively large investment and is a thorny issue for architectural conservation or maintenance [25]. Therefore, constant monitoring for decay and damage in traditional building structures is essential. During the investigation of the wooden frame structure, the situation was often urgent due to the presence of many problems that gradually emerged in the wooden structure over a long time, such as pillar base rotting, checking or splitting of the wood grain, parasite infestation and other kinds of damage to the wooden components that affect the building's overall stability. At present, the Hongcun villagers mostly ensure the traditional dwellings' temporary security and stability by increasing the lateral pillars or other supporting components (not always wooden components), which are not the practices most commonly recommended by modern conservation standards and cannot fundamentally solve the decay problem.

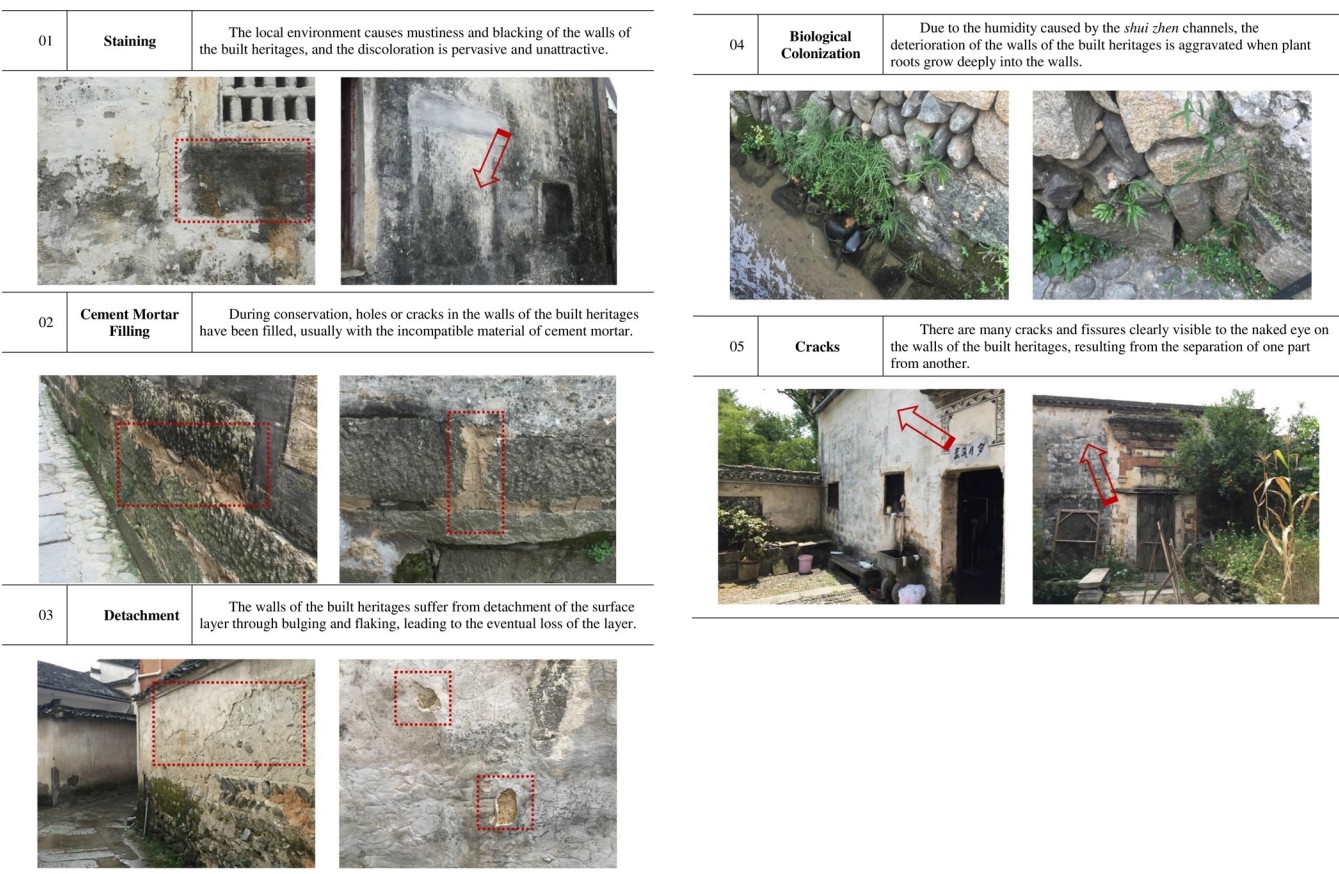

**Fig 4. Main decay phenomena of masonry components of traditional dwellings.**

The interior components of the traditional dwellings are made of wood as well and mainly include wooden windows and door frames, shutters and carved decorations, which are all interesting features of the traditional dwellings and the primary foci of conservation requirements. However, the skylight of a traditional dwelling is usually narrow and small; hence, the layout seriously affects lighting and ventilation in these buildings, negatively affecting humidity levels in the rainy season and causing interior wooden components to warp, thus reducing the stability of the structural members. Finally, insects are the most common problem with wooden components, especially with the humid climate, rich vegetation, and abundant rainfall in the Huizhou area. Insects devour wood, posing a threat to the structural stability and impacting the aesthetics of the buildings.

Based on the above analysis, the main decay phenomena of the wooden components included Knots, Color Change, Absence, Insect Attack, Bending, Cracks and Rot (see Fig 5). These were identified by referring to the relevant decay identification documents, and detailed text descriptions were developed in order by the level of damage to the structural stability of the Hongcun traditional dwellings and displayed through the specific decay photos of the wooden components.

## 5. Case study: Problematic conservation practices

The actual restoration projects of the traditional dwellings intuitively reflect the differences and problems arising from safeguard policies and conservation practices. During the

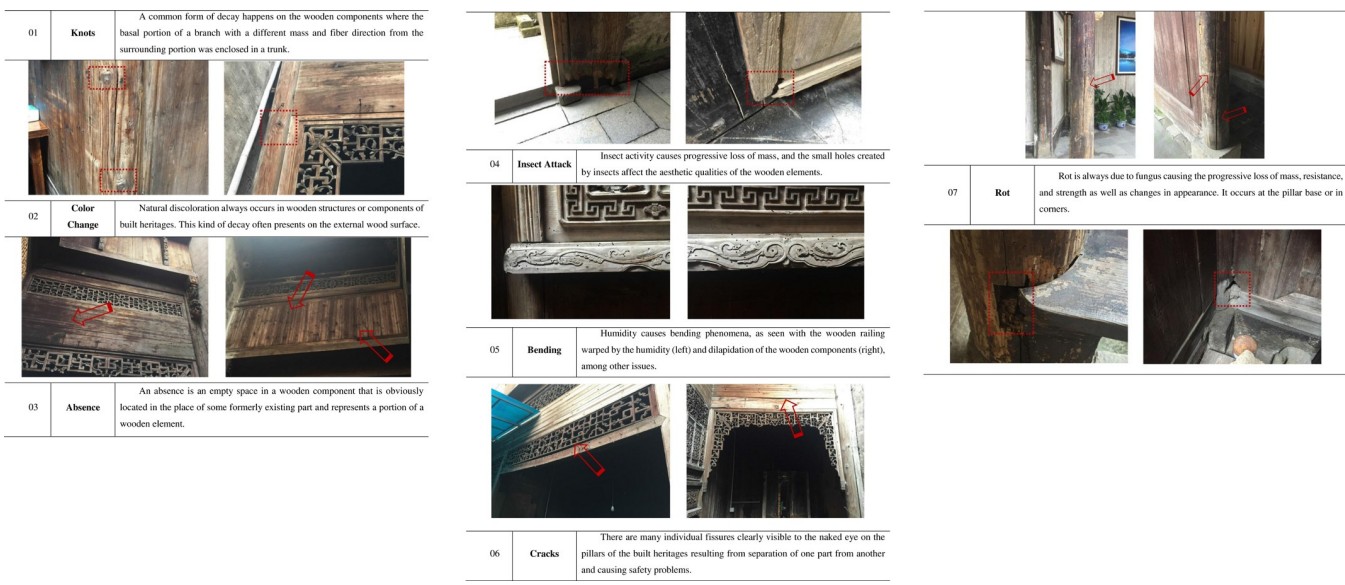

**Fig 5. Main decay phenomena of the wooden components of the traditional dwellings.**

investigation season in 2018 and 2019, several restoration projects for traditional dwellings were observed, and these were specifically divided into government-led projects and private projects. There is no doubt that Hongcun is a living heritage site, and modest renovation for public exhibition, for commerce use, or for self-residence is an effective means of extending the lives of the traditional dwellings. However, not only private projects but also government-led projects have conservation practices that are problematic in different ways. Therefore, specific examples of government-led and private projects will be examined in terms of actual conservation practices.

## 5.1 Case 1: The restoration project of Cheng Zhi Hall led by the local government

Cheng Zhi Hall is the largest traditional dwelling in Hongcun. It was built in the 5th year of the reign of Emperor Xianfeng of the Qing Dynasty (1855) and covers an area of 1639 m². The whole dwelling consists of 28 rooms, which are magnificent and glittering with exquisite carvings, red columns and golden beams; thus, it is honored as the *Civil Forbidden City*. In 1978, Cheng Zhi Hall was nationalized and is still open to the public as a major public exhibition spot.

Due to the enormous pressure of tourism, many components of Cheng Zhi Hall have been damaged to varying degrees. Therefore, to ensure the sustainability of Cheng Zhi Hall as a tourist site, in December 2015, the Yi County Cultural Relics Bureau commissioned the Anhui Provincial Cultural Relics Protection Center to complete the *Restoration Plan of Cheng Zhi Hall, Hongcun Village, Yi County, Huangshan City, Anhui Province* [26] and obtain approval and funding from the State Bureau of Cultural Relics. The restoration project began in February 2017 and finished in August 2018, and was carried out mainly according to the restoration plan. Consequently, the actual conservation problems could be identified by comparing the restoration plan with the actual restoration effects.

The restoration plan of Cheng Zhi Hall consists of six chapters: *Chapter 1: Restoration Instruction, Chapter 2: Photos of Current Status, Chapter 3: Survey Drawings, Chapter 4: Drawing of Restoration Plan, Chapter 5: Project Budget* and *Chapter 6: Insect and Ant Prevention*.

Following the relevant restoration principles, 241 photos of the current situation are available, as well as the 85 survey drawings (including the floor plans, elevations, sections and details of the components). According to the photographs of the actual situation and the survey drawings, the restoration plan is presented with the relevant restoration drawings, and construction detail drawings and puts forward corresponding restoration measures based on the specific problems of Cheng Zhi Hall. There are a total of 50 detail drawings marking the restoration measures on the corresponding plan and section drawings. However, as seen in these restoration drawings, the restoration measures listed are too general to conduct the conservation practices. For example, to intervene in the decay of walls, the restoration plan is described as *Chip away the bulging, musty and blackening lime plaster*, *rinse and re-plaster*. *Demolish the nontraditional walls and seek out and engage folk brick craftsmen to rebuild the walls*. This means there are no special solutions for how to restore the different decays of Cheng Zhi Hall.

Although the government made efforts to improve the situation of Cheng Zhi Hall through its restoration project, the actual situation is counterproductive due to different interpretations of the restoration principles and limitations in technical cognition, and many restoration results are debatable. In particular, the conservation practices one-sidedly emphasize *Identifiability* while neglecting *Compatibility*. For instance, the restoration project replaced the damaged wooden components with new components directly, destroying the original architectural value and features. In addition, although the restoration plan includes a description of the restoration instructions as well as status photos, survey drawings and restoration drawings, the most important content concerning the restoration measures of the architectural components are described only in very general words such as *Follow the existing size*, *quality*, *shape and original practice*, which does not involve comprehensive analysis and includes no practical measures at all. Due to the lack of scientific standards and measures, the conservation practices employed for Cheng Zhi Hall did not solve the actual decay phenomena and even caused a certain degree of damage to the architectural value and features of the hall (see Fig 6).

## 5.2 Case 2: The private project of restoring the Zha Songlin residence

For private restoration projects, because the restoration was funded primarily by the household, the government could not exercise strict supervision of the restoration project. As a result, the current state of the restoration presents several problematic issues.

The Zha Songlin Residence, formerly known as Shu Zi Hall, constructed in the Qing Dynasty (approximately 1850), was chosen as the case to analyze the typical phenomena occurring in private projects, as this traditional dwelling was under restoration during the investigation. The Zha Songlin Residence has been used continually as a private residence. In the latest Conservation Plan 2016, this building is identified under "*Buildings of the Cultural Relics Level*", and the preservation status is identified as "*Poor*", requiring "*Major Restoration*". As indicated above, the *Major Restoration* involves an investigation to determine the scale of the restoration, then replacing the damaged architectural components and correcting for elements unreasonably added or removed during previous restoration projects. Before the implementation of such restoration projects, in addition to meeting the requirements of the technical measures, relevant historical data and on-site technical research must be conducted to establish a basis for the preparation of a rigorous and scientific restoration plan. This must be reported to the relevant departments before approval of the implementation of the project. The Zha Songlin household commissioned the architectural conservation institution to compile the restoration plan, and it was approved by the Yi County World Cultural Heritage Management Office in 2017.

The restoration plan is called the *Design Plan for Emergency Reinforcement Project of Zha Songlin Residence*, *Hongcun Village*, *Yi County* [27] and is a complete conservation document

| 01 | **Actual Conservation Practices**: The Fei Zhao is one of the components of Chinese traditional architecture. It is usually a wood or carved panel made by hollowing, embossing, engraving and other techniques. The original Fei Zhao of the palanquin hall was missing. During the restoration, a newly made Fei Zhao was installed. However, the material was not carefully selected and does not have good compatibility with the original wooden components in terms of color and materials. |

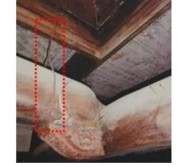
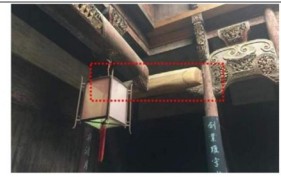

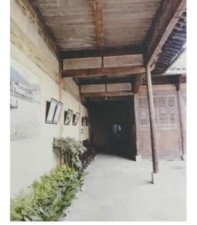
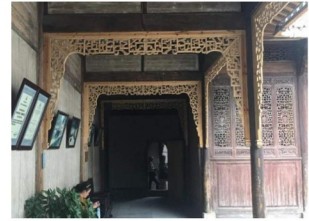

| 02 | **Actual Conservation Practices**: The front Fei Zhao of the palanquin hall was missing and was replaced with a newly made one. However, the material was again not carefully selected and does not have good compatibility with the original wooden components. Because the new front Fei Zhao of the palanquin hall is the main architectural component of Cheng Zhi Hall, the integrity of the hall's architectural style was destroyed. |

| 04 | **Actual Conservation Practices**: Before restoration, the grille door and the wooden railing of the second floor were missing, and the wooden window was partially damaged. However, after the restoration, the grille door and wooden railing were replaced with incompatible material, and the missing part of the wooden window was patched, damaging the original features of these parts of Cheng Zhi Hall. |

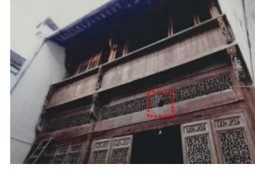
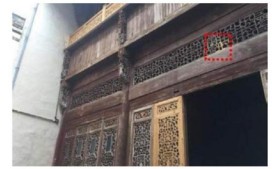

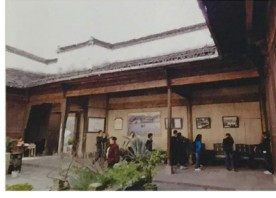
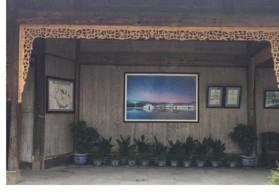

| 05 | **Actual Conservation Practices**: Before restoration, the walls were affected by mildew and blackening, and the plaster was spalling. However, during the restoration, the intervention measure was just to cover the damage with plaster, which was not compatible with the original wall. An appropriate process and compatible means, such as a germicidal reagent, were not used. |

| 03 | **Actual Conservation Practices**: The Mu Tuo Ni is the local name for Huizhou traditional dwellings but actually refers to the small beam supporting the main beam. Before restoration, the Mu Tuo Ni was rotten, and to ensure safety, the beam was reinforced with iron wire. However, after restoration, the Mu Tuo Ni was directly replaced. Actually, the original Mu Tuo Ni was still of good quality and just needed the right intervention instead of complete replacement. |

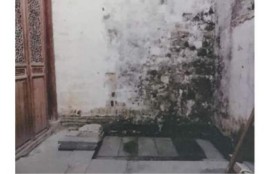
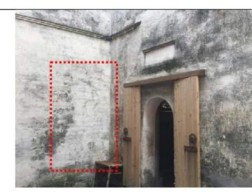

**Fig 6. Examples of the actual conservation practices applied to Cheng Zhi Hall.**

but does not mention the renovation content. In Chapter 1, the "*Plan Description*" is the main textual description of this plan. Chapter 2 provides the "*Status Photos*", with 56 photos of the residence's status in 2015 and a brief description of every photo detailing the actual problems with this traditional dwelling. Chapter 3 is the "*Survey Drawings*", which indicate the decay phenomena in 25 drawings of the plans, façade, sections and component details. Chapter 4 is the "*Restoration Drawings*", and it includes 12 drawings, with the restoration measures indicated on the relevant drawings. However, the drawings revealed very general and limited information, with only a simple sentence to indicate the conservation measures.

Setting aside the deficiencies of and problems with the restoration plan, the actual restoration practices applied to the Zha Songlin Residence were totally inconsistent with the plan. Instead, the project involved total renovation for eventual commercial use. Except for some obvious damage to the wooden structures that was repaired, the remaining restoration measures were not implemented. Rather, the original front hall of the Zha Songlin Residence, which was previously used as the living room, was renovated as a souvenir shop space. In addition, the courtyard of this traditional dwelling was renovated into a coffee shop with new construction and modern decorations, and even the façade of the Zha Songlin Residence was given a new window for selling takeaway beverages, which is prohibited in all versions of the Hongcun conservation planning and management policies (see Fig 7).

This so-called restoration project of the Zha Songlin Residence represents a clear violation of conservation planning and management policies. When the director of the *Law-enforcement Team of Hongcun*, whose responsibilities include the restoration supervision of the traditional dwellings, was asked about the restoration of Zha Songlin Residence, the response was *Our current management responsibility is to ensure the façade of the dwellings maintains the original*

| 01 | **Actual Conservation Practices**: The original front hall of the Zha Songlin Residence, which was used as the living room. However, after the intervention, it became the souvenir shop. |
|---|---|

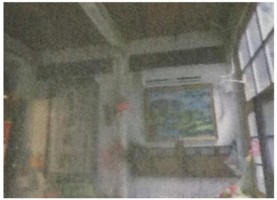 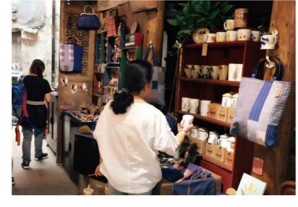 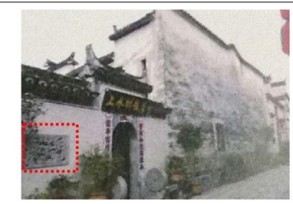 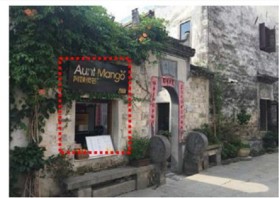

| 02 | **Actual Conservation Practices**: The courtyard of the Zha Songlin Residence was used to store debris before the intervention, but with the renovation, a new coffee shop was constructed that was not part of the restoration plan. |
|---|---|

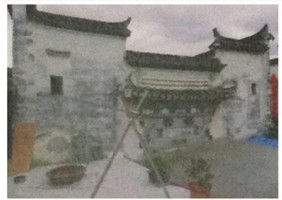 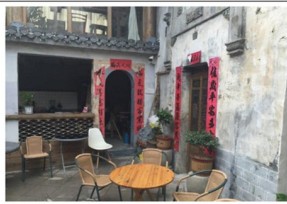

| 03 | **Actual Conservation Practices**: The façade of the Zha Songlin Residence now has a window for selling takeaway beverages. |
|---|---|

**Fig 7. Examples of the conservation practices applied to the Zha Songlin residence.**

*pattern and the height of the house is not changed. As for interior transformation, we cannot control it.* However, regarding the newly constructed window in the façade of Zha Songlin Residence, the director had no response. In fact, according to the provisions of the Conservation Plan 2016, cultural relics-level traditional dwellings can only be restored and continue their previous function; thus, the Zha Songlin Residence cannot be renovated for commercial use based on the regulation.

During the investigation, it became obvious that many private restoration projects followed this strategy of applying for restoration planning permission and then, once approval is given by the local government, proceeding "at will" during the actual "conservation" process. Coupled with the misunderstandings within the local management department regarding conservation planning and local management policies, as well as the inconsistent work attitude and the one-sided understanding of the renovation of the interior versus the façade, a considerable number of traditional dwellings have effectively been lost to the false "conservation" process. It is, therefore, essential to devise a methodology for establishing and implementing effective restoration standards and to implement a functional supervisory system. These factors are critical to guarantee the scientific conservation and restoration of traditional dwellings.

## 6. Strategies to optimize conservation of Hongcun traditional dwellings

From the situation outlined above, it may be understood that there are many deficiencies in Hongcun's conservation practices, necessitating discussion and the timely adjustment of conservation policies and their implementation. What needs to be clear is that only restoration and consolidation are allowed, with no alteration of existing volumes, and that traditional dwellings should not be jeopardized. Therefore, it can be explicitly seen that the rich historical information contained in differently built heritages and the conservation subjects should first

be clearly identified and understood, which is precisely what is obviously missing in Hongcun's biased conservation practices.

What, then, are the reasons for the large differences between the conservation practices and policies in Hongcun? Based on the analysis of the actual problems mentioned above, a lack of comprehensive surveys and knowledge, systematic conservation measures and scientific reuse criteria for traditional dwellings are the major problems. Therefore, if the conservation practices for traditional dwellings in Hongcun are to be effectively enhanced, there is an urgent need to develop more practical solutions to these issues, and the provisional solutions and suggestions are discussed in the following sections.

## 6.1 Comprehensive surveys of materials and decay of Hongcun traditional dwellings

To identify the most suitable conservation practices, the first step should be to gather a great deal of information on all the traditional dwellings within Hongcun. The survey should begin with a rigorous and thorough campaign to observe all the traditional dwellings and to successively move from general information to particular buildings' present status and decay, which is a scientific, step-by-step process.

Technical sheets can be used to maintain information on the traditional dwellings, including general information on the site, morphology, construction details (materials and construction techniques used for the architectural structure, walls, roofing, façade, ground and pavement, wooden components and decoration as well as technical equipment, etc.), current conservation status of the site and architectural components (mainly including the photographic records of the main decay phenomena of each architectural component and the priorities for the intervention measures in the next step);

In the second step of the survey campaign, the transferred technical sheet (cards, data and materials) should be analyzed and rearranged based on the information collected in the first step, primarily with reference to the building age and the typology of the traditional dwellings, in order to create a meaningful summary of the different types and the decay phenomena of the differently architectural components, illustrated by numerous and persuasive photographs and supported by brief descriptions.

## 6.2 Conservation measures for decay problems of Hongcun traditional dwellings

As discussed above, beginning with the earliest *Conservation Plan 1989*, Hongcun has been formulating all policies under the principle of authenticity and integrity, but there have never been specific criteria, categories or conservation measures to preserve the authenticity and integrity of traditional dwellings. Strictly speaking, the content of the conservation measures in all the Hongcun Conservation Plans seems to rely on interpretation of the nouns and does not solve any practical problems. In fact, Hongcun's conservation practices should be enhanced in two areas: conservation planning and specific restoration planning for single dwellings.

First, regarding conservation planning in Hongcun, based on fundamental knowledge, multiple alternative conservation practices should be proposed and should clearly outline which measures are incompatible with guaranteeing the least impact on traditional dwellings. Various solutions should be sought in the organization of conservation measures in conservation planning, such as attaching the "*Guidelines for Intervention Techniques*", which can help designers and workers form a more comprehensive understanding of the conservation and restoration of traditional dwellings.

Regarding the conservation of individual buildings, restoration planning should include *research and historical analysis*, *analysis of the actual condition*, *diagnosis of decay phenomena*, *conservation design* and the optional *reuse, development, management and maintenance design*, and the content should be as comprehensive as possible to provide substantial information and propose proper conservation measures for traditional dwellings. At the same time, based on the detailed analysis of decay phenomena, including a series of technical reports and drawings describing the deterioration and its causes, such as the distribution drawing of decay on a building, the proposed diagnostic opinions and conservation measures will be able to solve specific decay problems more effectively.

## 6.3 Reasonable criteria for renovation and reuse of Hongcun traditional dwellings

Scientific and rational reuse is a reasonable way to extend the life of the traditional dwellings of Hongcun, but the utilization intention should be proper, and reuse must occur within a reasonable framework with strict reuse criteria and a supervision mechanism to ensure the scientific preservation of traditional dwellings.

First, the traditional residential function and facilities of the traditional dwellings are no longer compatible with the needs of modern life. Consequently, the reuse criteria to be adopted for new usages need to be reconsidered. Because of the time-sensitive nature of this issue, the reuse criteria need to be defined clearly for all of Hongcun traditional dwellings, and the priority should be to ensure that the renovations maximize the conservation of the traditional construction techniques, the original layout of the dwelling, the traditional materials and the traditional components, with the aim of securing the permanence of the materials of both the exterior and interior.

Second, it is unrealistic to attempt to introduce new tourism modes rapidly, and doing so may cause more serious and uncontrolled damage. Renovation into hotels will likely not cease, and it is necessary to determine how to provide better reuse criteria under such situations. Therefore, there is a need to extensively discuss the many new functions that will be inserted into traditional dwellings to increase their intrinsic capacity, to maximize their vitality and to ensure their best usage.

## 7. Conclusion

In this study, the research process was highly logical and consistent. The materials and decay phenomena of Hongcun traditional dwellings were comprehensively investigated and described in detail in different sections. Afterwards, the problematic conservation practices for traditional dwellings, including government-led and private projects, were selected and examined in terms of actual conservation practices. Based on these analyses, the main problems related to the decay phenomena investigation and intervention were systematically summarized, and corresponding solutions were proposed to ensure the optimization of conservation strategies for the traditional dwellings in Hongcun village. The following conclusions were drawn:

1. The thorough investigation in this study led to a summary of the main building materials of Hongcun traditional dwellings as wooden structures supplemented by brick, tile and stone. The dwellings are made up of an exterior enclosure, a wooden structure and an interior component. Based on this, the existing decay phenomena of Hongcun traditional dwellings were identified and classified as Masonry Components and Wooden Components, which are listed in detail in the descriptions and photos.

2. Through the representative conservation cases of government-led and private projects, actual safeguard policies and conservation practices were observed and analyzed intuitively, and a series of practical problems were discovered and analyzed, including the unclear understanding of conservation principles, the incomplete investigation of decay phenomena, and the lack of conservation and restoration standards.

3. Based on these investigation and analysis results, optimization strategies were proposed to solve the deficiencies in conservation practices for Hongcun traditional dwellings, including a comprehensive investigation of the building materials and the decay of those materials, pertinent conservation measures to solve the decay problems, and reasonable renovation and reuse standards to ensure that conservation practices for Hongcun traditional dwellings are more scientific and effective in the future.

## Acknowledgments

We express our deepest gratitude to Prof. Maria Cristina Giambruno and Prof. Martin Sebastian Goffriller for their support during the preliminary research stages of this paper.

## Author Contributions

**Data curation:** Xianguang Gu.

**Supervision:** Xianguang Gu.

**Writing – original draft:** Tingshen Li.

**Writing – review & editing:** Minghao Zhang.

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
