## [Editor Report · Decision Letter 0]

27 Jul 2021

PONE-D-21-22843

On the Subject of the Decay Phenomena of and Conservation Practices Applied to Traditional Dwellings in Hongcun Village, China

PLOS ONE

Dear Dr. Tingshen,

Thank you for submitting your manuscript to PLOS ONE. After careful consideration, we feel that it has merit but does not fully meet PLOS ONE’s publication criteria as it currently stands. Therefore, we invite you to submit a revised version of the manuscript that addresses the points raised during the review process.

We look forward to receiving your revised manuscript.

Kind regards,

Mehmet Serkan Kirgiz

Academic Editor

PLOS ONE

Journal Requirements:

2. In your manuscript, please provide additional information regarding the specimens used in your study. Ensure that you have reported specimen numbers and complete repository information, including museum name and geographic location.

For more information on PLOS ONE's requirements for paleontology and archaeology research, see https://journals.plos.org/plosone/s/submission-guidelines#loc-paleontology-and-archaeology-research.

"NO authors have competing interests"

"This paper is funded from the CUMT Fundamental Research Funds for the Central Universities (No.

2020SK17)."

"CUMT Fundamental Research Funds for the Central Universities

Grant Number: 2020SK17"

7. We note that Figure 1 in your submission contain [map/satellite] images which may be copyrighted. All PLOS content is published under the Creative Commons Attribution License (CC BY 4.0), which means that the manuscript, images, and Supporting Information files will be freely available online, and any third party is permitted to access, download, copy, distribute, and use these materials in any way, even commercially, with proper attribution. For these reasons, we cannot publish previously copyrighted maps or satellite images created using proprietary data, such as Google software (Google Maps, Street View, and Earth). For more information, see our copyright guidelines: http://journals.plos.org/plosone/s/licenses-and-copyright.

Additional Editor Comments:

The lack of physical, mechanical, and durability properties of material inhibited me to understand your significant study and also enforced me to find proper reviewers who could understand its important value. To be able to start its review process properly, please add such details below.

1- Materials properties used in the region's traditional constructions.

2- Were the materials applied for any conservation steps during building process or after.

3- Please add information for traditional constructions in terms of conservation, retrofitting, and any other maintaining and diagnosing process.

4- Such problems you have faced in your buildings are not private problems for your buildings. Almost all constructions faces such problems because of ground water, humidity, false construction techniques, ec. So, please tell us more about source of problems in your buildings.

After adding the details mentioned above, please resubmit it to the PLOS One.
---

## [Author Response · Author response to Decision Letter 0]

2 Mar 2022

Dear Dr.Oliver Ian Tolentino

On behalf of my co-authors, we thank you very much for giving us an opportunity to revise our manuscript, we appreciate very much for the positive and constructive comments and suggestions on our manuscript entitled “On the Subject of the Decay Phenomena of and Conservation Practices Applied to Traditional Dwellings in Hongcun Village, China” (Manuscript Number: PONE-D-21-22843).

Those comments are all valuable and very helpful for revising and improving our paper, as well as the important guiding significance to our researches. We have studied comments carefully and have made correction which we hope meet with approval. Revised portion are marked in the manuscript. 

We would like to express our great appreciation to you for comments on our paper. Looking forward to hearing from you.

Thank you and best regards.

Yours sincerely,

Corresponding author:

Tingshen Li

litingshen1988@163.com

---

## [Decision Letter · Decision Letter 1]

24 Jun 2022

PONE-D-21-22843R1On the Subject of the Decay Phenomena of and Conservation Practices Applied to Traditional Dwellings in Hongcun Village, ChinaPLOS ONE

Dear Dr. Li,

Thank you for submitting your manuscript to PLOS ONE. After careful consideration, we feel that it has merit but does not fully meet PLOS ONE’s publication criteria as it currently stands. Therefore, we invite you to submit a revised version of the manuscript that addresses the points raised during the review process.

We look forward to receiving your revised manuscript.

Kind regards,

Mehmet Serkan Kirgiz PhD, MSc, BSc (He/Him/His)

Professor of Istanbul Sabahattin Zaim University 

Academic Editor of PLOS ONE

Journal Requirements:

Reviewers' comments:

Reviewer's Responses to Questions

**Comments to the Author**

1. If the authors have adequately addressed your comments raised in a previous round of review and you feel that this manuscript is now acceptable for publication, you may indicate that here to bypass the “Comments to the Author” section, enter your conflict of interest statement in the “Confidential to Editor” section, and submit your "Accept" recommendation.

Reviewer #1: (No Response)

Reviewer #2: (No Response)

2. Is the manuscript technically sound, and do the data support the conclusions?

Reviewer #1: Partly

Reviewer #2: (No Response)

3. Has the statistical analysis been performed appropriately and rigorously? 

Reviewer #1: No

Reviewer #2: (No Response)

4. Have the authors made all data underlying the findings in their manuscript fully available?

Reviewer #1: No

Reviewer #2: (No Response)

5. Is the manuscript presented in an intelligible fashion and written in standard English?

Reviewer #1: Yes

Reviewer #2: (No Response)

6. Review Comments to the Author

Reviewer #1: After reviewing the paper carefully, the authors performed most of the comments carefully. But the following comments should be addressed before publication:

1- The title is not clear and needs to be replaced by a more appropriate

2- Abstract should be re-written. The abstract should reflect the whole performed study clearly.

3- The whole introduction should be re-written completely. Only a few papers were referenced in the introduction section without talking about their results deeply.

4- The exact evaluation should be discussed in Introduction and also the previous assessed methods should be presented. In this area different studies have been done that necessary to present in Introduction as solutions: https://doi.org/10.1016/j.engstruct.2022.114358, https://doi.org/10.1016/j.istruc.2021.11.037, https://doi.org/10.1016/j.istruc.2021.08.089, https://doi.org/10.1016/j.engstruct.2021.112325, https://doi.org/10.1016/j.engstruct.2021.112122, https://doi.org/10.1016/j.engstruct.2020.111523.

5- Please avoid writing Chinese in English paper like 牵一发而动全身”

6- The conclusion section should be added to summarize all main obtained results from this study

7- Divide different structures and present their information step by step: 1- location, 2- materials, 3- application, 4- problem and finally 5- your suggested solution

Based on those represented above, this paper should be considered for another review after major revision.

Reviewer #2: (No Response)

7. PLOS authors have the option to publish the peer review history of their article (what does this mean?). If published, this will include your full peer review and any attached files.

Reviewer #1: **Yes: **Arash Karimi Pour

Reviewer #2: No

---

## [Author Response · Author response to Decision Letter 1]

23 Jul 2022

Dear reviewers,

On behalf of my coauthors, we thank you very much for giving us an opportunity to revise our manuscript. We appreciate the positive and constructive comments and suggestions on our manuscript Optimization Strategies for Conservation of Traditional Dwellings in Hongcun Village, China, Based on Decay Phenomena Analysis (Number: PONE-D-21-22843).

First, I would like to mention that Tingshen Li’s affiliation has changed to College of Environmental Engineering, Xuzhou University of Technology, Xuzhou 221018, China, and the corresponding author has been changed to Prof. Minghao Zhang. At the same time, the funding source has also changed to the Xuzhou Science and Technology Plan Project (No. KC21305). We apologize for any inconvenience to the journal.

The comments provided were all valuable and very helpful for revising and improving our paper and providing important guidance to our research. We have studied the comments carefully and have made corrections that we hope will meet with your approval. The revised portions are marked in the manuscript. The main corrections in the manuscript and the responses to the editors’ comments are as follows:

Reviewer 1-1. The title is not clear and needs to be replaced by a more appropriate.

Response: After comprehensively analyzing the full research, we revised the title of the paper to " Optimization Strategies for Conservation of Traditional Dwellings in Hongcun Village, China, Based on Decay Phenomena Analysis" in order to indicate the emphasis of this study.

Reviewer 1-2. Abstract should be re-written and reflect the whole performed study clearly.

Response: The abstract has been completely rewritten, and the whole research process is clearly introduced according to the logical structure of the Material and Decay Investigation, the Problematic Conservation Cases Analysis, and the Conservation Measures Optimization Strategies.

Reviewer 1-3-4. The whole introduction should be re-written completely. Only a few papers were referenced in the introduction section without talking about their results deeply. The exact evaluation should be discussed in Introduction and also the previous assessed methods should be presented. In this area different studies have been done that necessary to present in Introduction as solution

Response: The introduction was completely rewritten by adding the results of many previous studies regarding building materials, decay phenomena, and conservation measures for Hongcun traditional dwellings. Additionally, the Research Significance and Methods section was added to elaborate and summarize the deficiencies of previous research and to discuss the specific research methods and procedures of this study.

Reviewer 1-5. Please avoid writing Chinese in English paper

Response: All Chinese characters in the article have been deleted.

Reviewer 1-6. The conclusion section should be added to summarize all main obtained results

Response: The Conclusion section has been added to summarize all the main results in the three aspects of the Material and Decay Investigation, the Problematic Conservation Cases Analysis, and the Conservation Measures Optimization Strategies.

Reviewer 1-7. Divide different structures and present their information step by step: 1- location, 2- materials, 3- application, 4- problem and finally 5- your suggested solution

Response: The overall structure of this study has been reclassified as the general introduction (including the general information about and location of Hongcun), the survey and summary of the main Hongcun building materials, the investigation and analysis of the existing decay phenomena, the problematic conservation cases and finally the strategies for optimizing conservation practices. The revised structure of this paper has a clearer overall logic.

Reviewer 2-1. The research works that were performed in this field in recent years (after 2016) are not mentioned. Please cite more references

Response: More research published after 2017 has been cited, especially in the rewritten Introduction, to further introduce the trends of research in this field.

Reviewer 2-2. Please provide an overview of existing problems and the solutions and suggestions needed to address them in form of a flowchart

Response: A research flowchart has been added to show the whole research process, existing problems and potential solutions in detail.

Reviewer 2-3. Lack of a general "Conclusion" section after which suggestions are made

Response: The Conclusion section has been added to summarize all the main results in the three aspects of the Material and Decay Investigation, the Problematic Conservation Cases Analysis, and the Conservation Measures Optimization Strategies.

We are greatly appreciative of your comments on our paper. We look forward to hearing from you.

Thank you and best regards.

Yours sincerely,

Minghao Zhang

407255050@qq.com

---

## [Decision Letter · Decision Letter 2]

5 Oct 2022

Optimization Strategies for Conservation of Traditional Dwellings in Hongcun Village, China, Based on Decay Phenomena Analysis

PONE-D-21-22843R2

Dear Dr. ZHANG,

We’re pleased to inform you that your manuscript has been judged scientifically suitable for publication and will be formally accepted for publication once it meets all outstanding technical requirements.

Kind regards,

Nazanin Tajik

Academic Editor

PLOS ONE

Additional Editor Comments (optional):

Reviewers' comments:

Reviewer's Responses to Questions

**Comments to the Author**

1. If the authors have adequately addressed your comments raised in a previous round of review and you feel that this manuscript is now acceptable for publication, you may indicate that here to bypass the “Comments to the Author” section, enter your conflict of interest statement in the “Confidential to Editor” section, and submit your "Accept" recommendation.

Reviewer #1: All comments have been addressed

2. Is the manuscript technically sound, and do the data support the conclusions?

Reviewer #1: Yes

3. Has the statistical analysis been performed appropriately and rigorously? 

Reviewer #1: Yes

4. Have the authors made all data underlying the findings in their manuscript fully available?

Reviewer #1: Yes

5. Is the manuscript presented in an intelligible fashion and written in standard English?

Reviewer #1: Yes

6. Review Comments to the Author

Reviewer #1: The authors are appreciated for addressing all comments carefully. So, the paper is recommended for publication in the current format

7. PLOS authors have the option to publish the peer review history of their article (what does this mean?). If published, this will include your full peer review and any attached files.

Reviewer #1: **Yes: **Arash Karimi Pour

---

## [Editor Report · Acceptance letter]

13 Oct 2022

PONE-D-21-22843R2 

Optimization Strategies for Conservation of Traditional Dwellings in Hongcun Village, China, Based on Decay Phenomena Analysis 

Dear Dr. ZHANG:

I'm pleased to inform you that your manuscript has been deemed suitable for publication in PLOS ONE. Congratulations! Your manuscript is now with our production department. 

Kind regards, 

on behalf of

Dr. Nazanin Tajik 

Academic Editor

PLOS ONE